# Age and Sex-Related Effects on Single-Subject Gray Matter Networks in Healthy Participants

**DOI:** 10.3390/jpm13030419

**Published:** 2023-02-26

**Authors:** Yoko Shigemoto, Noriko Sato, Norihide Maikusa, Daichi Sone, Miho Ota, Yukio Kimura, Emiko Chiba, Kyoji Okita, Tensho Yamao, Moto Nakaya, Hiroyuki Maki, Elly Arizono, Hiroshi Matsuda

**Affiliations:** 1Department of Radiology, National Center Hospital, National Center of Neurology and Psychiatry, Tokyo 187-8551, Japan; 2Center for Evolutionary Cognitive Sciences, Graduate School of Art and Sciences, The University of Tokyo, Tokyo 153-8902, Japan; 3Department of Psychiatry, Jikei University School of Medicine, Tokyo 105-8461, Japan; 4Department of Neuropsychiatry, University of Tsukuba, Tsukuba 305-8576, Japan; 5Department of Drug Dependence Research, National Institute of Mental Health, National Center of Neurology and Psychiatry, Tokyo 187-8551, Japan; 6Department of Psychiatry, National Center Hospital, National Center of Neurology and Psychiatry, Tokyo 187-8551, Japan; 7Department of Radiological Sciences, School of Health Sciences, Fukushima Medical University, Fukushima 960-8516, Japan; 8Department of Radiology, Juntendo University School of Medicine, Tokyo 113-8421, Japan; 9Department of Radiology, Graduate School of Medicine, The University of Tokyo, Tokyo 113-8655, Japan; 10Department of Biofunctional Imaging, Fukushima Medical University, Fukushima 960-1295, Japan; 11Drug Discovery and Cyclotron Research Center, Southern Tohoku Research Institute for Neuroscience, Fukushima 963-8052, Japan

**Keywords:** aging, sex, gray matter, magnetic resonance imaging, network

## Abstract

Recent developments in image analysis have enabled an individual’s brain network to be evaluated and brain age to be predicted from gray matter images. Our study aimed to investigate the effects of age and sex on single-subject gray matter networks using a large sample of healthy participants. We recruited 812 healthy individuals (59.3 ± 14.0 years, 407 females, and 405 males) who underwent three-dimensional T1-weighted magnetic resonance imaging. Similarity-based gray matter networks were constructed, and the following network properties were calculated: normalized clustering, normalized path length, and small-world coefficients. The predicted brain age was computed using a support-vector regression model. We evaluated the network alterations related to age and sex. Additionally, we examined the correlations between the network properties and predicted brain age and compared them with the correlations between the network properties and chronological age. The brain network retained efficient small-world properties regardless of age; however, reduced small-world properties were observed with advancing age. Although women exhibited higher network properties than men and similar age-related network declines as men in the subjects aged < 70 years, faster age-related network declines were observed in women, leading to no differences in sex among the participants aged ≥ 70 years. Brain age correlated well with network properties compared to chronological age in participants aged ≥ 70 years. Although the brain network retained small-world properties, it moved towards randomized networks with aging. Faster age-related network disruptions in women were observed than in men among the elderly. Our findings provide new insights into network alterations underlying aging.

## 1. Introduction

The brain undergoes several structural and functional alterations associated with normal aging. Magnetic resonance imaging (MRI) is a robust tool for evaluating the alterations in the brain volume in vivo. Previous studies have demonstrated the effects of aging and sex on gray matter volume [1,2,3], cortical thickness [4], and white matter volume [1,2]. The voxel-based morphometry analysis of 563 healthy subjects (age range: 20–86, female 55%) revealed linear age-related declines of normalized gray matter volume with advancing age [3]. The nonlinear age-related declines of normalized white matter volume (i.e., an inverted U-shape) with slightly increased volume during adulthood, which reaches a peak in the fourth decade, is consistent with the ongoing maturation of the white matter [1,3]. Regarding the effects of sex, larger normalized gray matter in females compared to males and steeper age-related decline in females compared to males were reported in both younger (142 subjects, age range: 20–34, female 50%) and older groups (135 subjects, age range: 60–86, female 51%) [2]. 

Recent advances in image analyses have made investigating brain networks based on graph theory [5] and predicting brain age using machine learning possible [6]. Brain networks can be assessed using several modalities, including resting-state functional magnetic MRI, diffusion tensor imaging, and three-dimensional (3D) T1-weighted images. Among them, 3DT1-weighted images have the advantage of obtaining stable images with a short acquisition time, which are not affected by physiological conditions. Although previous network studies using 3DT1-weighted images are limited to inter-group analysis of cortical thickness or gray matter volume across patients [7,8], a single-subject cortical similarity-based analysis enabled the assessment of network alterations at the individual level [9]. Briefly, an individual graph was assessed by examining statistically similar gray matter morphology between small brain regions within a single participant. It has become possible to observe network trajectories related to age based on the individual’s network results rather than comparing different age groups. Since network properties are linked to cognitive function, this network analysis might provide new information reflecting an individual’s brain function. Thus, the careful observation of an individual’s network may enable the prediction of future disease risk. This analysis has been used in neuropsychiatric diseases such as Alzheimer’s disease [10,11], epilepsy [12], and bipolar disorder [13]. However, no studies have examined age- and sex-related network alterations using similarity-based analyses in healthy participants. 

The prediction of an individual’s brain age is also possible using gray matter images [6]. Machine learning is employed to study the pattern of the image data of a large number of healthy participants; when the individual’s data are inputted into the model, the brain age of the given individual can be predicted. Brain age has been used in several diseases, including Alzheimer’s disease [6,14], Parkinson’s disease [14], and epilepsy [15], and is considered a useful biomarker for detecting and monitoring neurodegenerative disorders. 

We hypothesized that the brain network declines with age, and we also hypothesized that sex differences could affect age-related network alterations considering hormonal changes and a higher incidence of Alzheimer’s disease (typical age-related neurodegenerative disorders) in women than in men. Furthermore, we speculated that predicted brain age correlates better with network properties than chronological age. In this study, we calculated small-world network properties using single-subject gray matter network analysis and investigated network alterations related to age and sex in a large sample of healthy participants.

## 2. Materials and Methods

### 2.1. Participants

We employed 812 MRI scans (407 females and 405 males; age, mean ± standard deviation, 59.3 ± 14.0 years, range 25.8–85.1 years) from the normal database at our center. The age distribution was even for females and males in all generations. The age distributions of the participants used in this study are shown in Figure 1.

All the participants were Japanese and healthy, with no history of neurological or psychiatric disorders and no history of use of medication that affects the central nervous system, according to the medical interviews. All the participants underwent 3D sagittal T1-weighted imaging. Visual inspection revealed no structural abnormalities or significant artifacts. Written informed consent was obtained from all the participants. This study was approved by the Institutional Review Board of the National Center of Neurology and Psychiatry and was performed in accordance with the Declaration of Helsinki.

### 2.2. MRI Data Acquisition 

All 3D sagittal T1-weighted images were obtained using two 3-T MRI scanners with 32-channel head-neck coils. A total of 489 MRI scans were acquired using a Philips 3-T MRI scanner (Philips Medical Systems, Best, The Netherlands) with the following protocol: repetition time (TR)/echo time (TE), 7.18/3.46 ms; flip angle (FA), 10°; number of excitations (NEX), 1; 0.68 × 0.68 mm^2^ in-plane resolution; matrix, 384 × 384; field of view (FOV), 26.1 × 26.1 cm; 0.6 mm effective slice thickness with no gap; 300 slices; and acquisition time, 4 min 4 s. The other 323 MRI scans were acquired using a Siemens 3-T MRI scanner (Verio, Siemens, Erlangen, Germany) with the following protocol: TR/TE, 1800/2.25 ms; FA 9°; NEX, 1; 0.87 × 0.78 mm^2^ in-plane resolution; matrix, 320 × 280; FOV, 25 × 25 cm; 0.8 mm effective slice thickness with no gap; 224 slices; and acquisition time, 5 min 27 s. 

### 2.3. MR Data Preprocessing and Brain Age Prediction

The 3D T1-weighed images were preprocessed using Statistical Parametric Mapping 12 (SPM12; Functional Imaging Laboratory, University College London, London, UK) running on MATLAB 2021b (Mathworks, Natick, MA, USA). The gray matter images were segmented and spatially normalized using the DARTEL (anatomical registration of differential morphology with exponential Rye algebra) algorithm implemented in the Computational Anatomy Toolbox (CAT 12). Finally, the normalized gray matter images were smoothed using a 4 mm full width at half maximum (FWHM) Gaussian kernel for brain age prediction [15] and an 8 mm FWHM Gaussian kernel for similarity-based gray matter network analysis [10]. 

### 2.4. Brain Age Prediction 

To predict the brain age, we used the support regression model implemented in the LIBSVM (http://www.csie.ntu.edu.tw/~cjlin/libsvm/) toolbox with a linear kernel and default setting (i.e., in the LIBSVM: C = 1, v = 0.5). Principal component analysis was used to reduce the overfitting and overcome the curse of dimensionality, and the number of principal components was set to 100 as previously described [15]. For the regression model, the principal components derived from gray matter voxel intensities were considered independent variables, and the chronological age was considered a dependent variable. To evaluate the ability of the regression model, we used 10-fold cross-validation, with one fold in iteration considered as the test and the remaining folds being fitted to the model. The accuracy of the model was measured by using the mean absolute error, root-mean-square error, and the correlation between chronological age and predicted brain age through 10-fold cross-validation. Finally, differences in the predicted brain and chronological ages (brain-PAD: predicted age-chronological age) were calculated for each participant. 

### 2.5. Single-Subject Gray Matter Network

Single-subject gray matter networks were computed using a previously described automated pipeline (http://github.com/bettytijms/Single_Subject_Grey_Matter_Networks; version 20,150,902 accessed on 3 May 2019) [9]. Figure 2 shows a diagram of the proposed method. A brain network is defined as the nodes and edges that connect them. In this study, nodes were defined as cubes of 3 × 3 × 3 voxels extracted from gray-matter images. Men had significantly more nodes than women (7091.90 ± 514.56 vs. 6378.06 ± 457.28, Mann–Whitney U test, *p* < 0.0001). Connectivity represents statistically high cortical similarities between any two nodes in a single participant. To identify the maximum similarities, each node was rotated by an angle *θ* with multiples of 45° and reflections over all the axes. Thereafter, to construct unweighted and undirected graphs, the similarity matrix was binarized with a threshold to include similarity values that reached a significance level of *p* < 0.05, corrected for multiple comparisons with false discovery rate (correlations greater than the threshold were indicated as 1, and correlations lower than the threshold were indicated as 0).

The most commonly used network properties are the clustering coefficient and path length. The clustering coefficient shows the tendency of the interconnectedness of neighboring nodes and is considered a measure of segregation. The path length is defined as the average short path between any two nodes and is considered a measure of integration. Based on these two properties, networks are classified into three types: regular (high clustering and long path length), random (low clustering and short path length), and small-world (high clustering and short path length), which lies between regular and random (Figure 3). In healthy individuals, the brain network is considered to maintain an efficient “small-world property [16].” To estimate whether a real brain network has a small-world topology, the clustering coefficient and path length in the real network is compared with those in a matched random network. Thus, we also computed five randomized networks for each binarized matrix.

Finally, we calculated the following small-world properties: normalized clustering (ratio of average clustering to that of its randomized version), normalized path length (ratio of average path length to that of its randomized version), and small-world coefficient (normalized clustering divided by normalized path length). The network is defined as a “small-world property” when normalized clustering > 1, normalized path length ≈ 1, and small-world coefficient > 1 [17]. 

### 2.6. Statistical Analysis

To evaluate the variations in sex in demographics, including chronological/predicted brain age, brain PAD score, number of nodes, and global network properties, we performed the Mann–Whitney U test. The statistical significance level was set at a *p*-value < 0.05 and at a *p*-value < 0.017 (=0.05/3) for network properties after Bonferroni correction.

Figure 4 shows scatter plots between network properties and age, controlling for the number of nodes and scanner types in women and men. The age trajectory curve was fitted using a generalized additive model (GAM). However, since the visual inspection of GAM revealed faster age-related declines in women aged ≥ 70 years, we considered a generalized linear model (GLM) to investigate the effects of age and sex on network properties. Thus, we divided the participants into two subgroups using a threshold of 70 years: <70 years old and ≥70 years old. First, we tested the interaction effect of age and sex on global network properties. If no interaction existed, we further tested the main effect of age and sex on network properties. To eliminate the effect of the number of nodes (men > female, Mann–Whitney U test, *p* < 10^−4^), which reflect gray matter volume, we included them as nuisance covariates. We also directly compared the sex differences using analysis of covariance, controlling for age, the number of nodes, and scanner types. Statistical analyses were performed using R version 4.2.1.

Additionally, the correlations between each network property and chronological/brain age were evaluated using partial correlation analysis controlling for sex, the number of nodes, and scanner types in the two subgroups. The statistical significance level was set at a *p*-value < 0.025 (=0.05/2) after the Bonferroni correction. Comparisons of correlations for each network property of chronological age and brain age were analyzed using Psychometrica [18], and a *p*-value < 0.05 was deemed significant.

## 3. Results

### 3.1. Demographics

The participants’ characteristics are presented in Table 1. There were no significant sex differences in the predicted brain age and brain-PAD score. Considering the network properties, both women and men retained a small-world topology showing normalized clustering > 1, normalized path length ≈ 1, and small-world coefficient > 1. Women exhibited significantly higher small-world properties (normalized clustering, normalized path length, and small-world coefficient) than men. 

### 3.2. Age- and Sex-Related Alterations of Network Properties

Table 2 shows the results of the effects of age and sex of the small-world network properties. In participants aged < 70 years, a significant interaction of age and sex was observed in the normalized path length. However, normalized clustering and small-world coefficients exhibited no interactions between age and sex, suggesting that age-related network alterations were similar in women and men. The main effects of age were observed in both groups, and an effect of sex was observed in normalized clustering.

In participants aged ≥ 70 years, a significant interaction of age and sex was observed on all small-world measures, suggesting that age-related network alterations vary between women and men. Age-related network decline was faster in women than in men. 

Direct comparisons of women and men revealed significantly higher network properties in women than in men in participants < 70 years old; however, these differences in sex disappeared in participants ≥ 70 years old (Table 3).

### 3.3. Correlations between Global Network Properties and Chronological/Brain Age

The correlation results between global network properties and chronological/brain age in patients aged < 70 and ≥70 years are shown in Table 4. For chronological/brain age < 70 years, the partial correlation test results showed moderate correlations between the network properties and chronological/brain age, and no significant correlations were found. In chronological/brain age ≥ 70 years, chronological age showed only mild correlations with the network properties; however, brain age showed significantly higher correlations than chronological age.

## 4. Discussion

To our knowledge, this is the first study to investigate age- and sex-related network alterations based on a single-subject gray matter network analysis in a large sample of healthy individuals. Although women exhibited higher network properties compared to men and similar age-related network declines as men in the participants aged < 70 years, faster age-related network declines over time were observed in women and led to no sex differences in the participants aged ≥ 70 years. Moreover, brain age was highly correlated with network properties compared to chronological age in the elderly. Our findings on age- and sex-related alterations in gray matter network properties may contribute to a better understanding of the mechanisms underlying normal aging.

Although the brain network retained small-world topology regardless of age, the small-world properties (normalized clustering, normalized path length, and small-world coefficient) declined with advancing age, indicating that the network moved towards a less optimal random organization. Considering normalized clustering, our findings were compatible with previous network studies using resting-state functional MRI in 15 healthy young (mean age = 24.7 years) and 11 healthy old adults (mean age = 66.5 years) [19] and diffusion tensor tractography in 95 healthy individuals aged 19–85 years [5], showing reduced local efficiency (corresponding to lower clustering [20]) in aging. Since clustering reflects local information transfer within small brain regions, the decreased clustering found in this study may suggest that local information transfer becomes less efficient with normal aging. However, inconsistent results were observed for the direction of normalized path length. We found reduced normalized path length in aging; however, previous studies using resting-state functional MRI [19] found results contrary to ours, showing reduced global efficiency (corresponding to longer path length [20]) in older adults than in young adults. Another network study using regional gray matter volume in 350 randomly selected healthy participants for each group from a large database reported higher global efficiency in the middle-aged group (41–60 years) than in young (18–40 years) and old age groups (61–80 years) [21]. These inconsistent results concerning the direction of path length could be attributed to the differences in the methodology used to construct the network or network parameters that are not computationally equivalent (i.e., path length vs. global efficiency). 

In Alzheimer’s disease, a typical age-related neurodegenerative disorder, gray matter networks are known to disrupt and move to random networks [11]. A recent study using single-subject network analysis revealed that gray matter networks develop a more random organized topology in individuals with subjective cognitive decline, which could be a very early preclinical stage of Alzheimer’s disease [22]. They found associations between lower normalized clustering and normalized path length with a longitudinal decline in global cognition. Another single-subject network study also reported the association of lower normalized clustering and small-world coefficient with increased risk of clinical progression in participants without dementia with abnormal amyloid cerebrospinal fluid markers [23]. A more recent study on autosomal dominant Alzheimer’s disease mutation carriers demonstrated that the earliest network difference relative to non-carriers was observed in a lower normalized path length, followed by lower normalized clustering and small-world coefficient, but not captured in other network parameters [24]. Previous single-subject network studies have consistently revealed that small-world properties are sensitive markers for detecting changes in gray-matter networks. In this study, we first demonstrated gray matter network trajectories by age using a large sample of the normal database. Although we cannot rule out the possibility that patients in the predementia stage were included in our sample, our findings of age-related lower small-world properties suggest that gray matter networks develop a more randomized topology even with normal aging. 

Considering the sex differences, women showed more optimal network topology showing higher small-world properties compared to men in participants aged < 70 years. 

Consistent with our results, previous weighted network studies using diffusion tensor tractography in 95 healthy individuals aged 19 to 85 years [5] and 72 young individuals aged 18 to 27 years [25] found higher local efficiency (corresponding to higher clustering [20] in unweighted networks) in women than in men; however, they found higher global efficiency (corresponding to shorter path length [20] in unweighted networks) in women than in men. Inconsistent results concerning the direction of path length could be owing to differences in the methodology used to construct the network (i.e., unweighted vs. weighted network). A larger corpus callosum in women [26] possibly enables greater interhemispheric information transfer, which might account for the longer path length in women observed in our study. 

There is accumulating evidence of the neuroprotective effect of estrogen against brain aging [27,28,29]. Estrogen therapy protects against apoptotic cell death by enhancing the expression of genes that optimize cell survival [30]. Moreover, a number of studies have clarified that hormone replacement therapy reduces the number and size of white matter hyperintensities in the brain [31]. Previous functional MRI studies have reported that ovarian hormones may enhance both cortico-cortical and subcortico-cortical functional connectivity [32,33]. Interestingly, we found faster age-related network declines in women than in men in participants aged ≥70 years. We hypothesized that hormonal changes would affect network alterations in women. However, network disruptions in individuals aged ≥70 years would not be accounted for only in the menopausal state since the average age of menopause is in the early to mid-50s [34]. Our findings of network decline at an older stage than menopause may account for the female advantages in verbal memory during normal cognitive aging [35]. This female advantage may act as a cognitive reserve and mask the early sign of cognitive decline despite comparable brain pathology to men; moreover, women show a faster cognitive function decline after brain pathology progression [36]. 

We also explored the correlations between the global network properties and chronological/predicted brain age. In the chronological/brain age ≥ 70 years, brain age was highly correlated with network properties compared to chronological age, which only showed a weak correlation. With age, metabolic diseases and the risk of neurodegenerative diseases are increasing. A recent study on cognitively unimpaired elderly individuals reported that diabetes and alcohol use are associated with older brain age [37]. They also reported that higher life satisfaction was associated with a younger brain age. Thus, several factors, including lifestyle change, affect brain age, and it would be appropriate to consider brain age when evaluating brain networks in the elderly.

This study had several limitations. First, we conducted a network analysis using only the 3DT1-weighted images. It is important to study age- and sex-related network changes using resting-state functional MRI and diffusion MRI and the associations of network alterations with the present study. Second, since this was a cross-sectional study, we could not confirm that the rapid age-related decline in elderly women would be a potential risk factor for cognitive decline or a predementia state of neurodegenerative disease. Thus, longitudinal network alterations will be warranted in the future. Third, for the prediction of brain age, we set the number of principal components = 100 using the same model as previously described [15]. Since the number of principal components is controversial, it may be desirable to determine the parameters using the Monte Carlo method in the future. Fourth, we have no inclusion or exclusion criteria for the risk factors of cardiovascular disease. Previous structural gray matter networks of 616 healthy elderly (age range: 60–80, female 42%) revealed that cardiovascular risk factors such as smoking, higher blood pressure, higher glucose, and higher visceral obesity were negatively associated with structural networks [38]. We have to keep in mind that these risk factors might affect the network declines observed in the elderly in this study. 

## 5. Conclusions

We used single-subject network analysis to investigate age- and sex-related network alterations in a large sample of healthy individuals. Although women exhibited higher network properties than men and similar age-related networks declined as men aged < 70 years, rapid age-related network declines were observed in women aged ≥ 70 years. Moreover, brain age was highly correlated with network properties compared to chronological age in the elderly. Our findings on age- and sex-related gray matter network alterations may contribute to a better understanding of the mechanisms underlying normal aging.

## Figures and Tables

**Figure 1 jpm-13-00419-f001:**
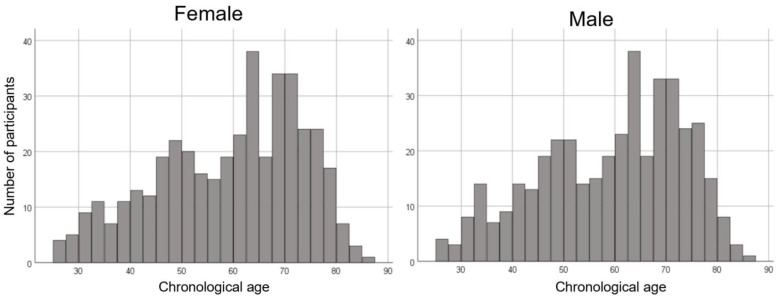
Histogram showing the age distribution for females and males in this study.

**Figure 2 jpm-13-00419-f002:**
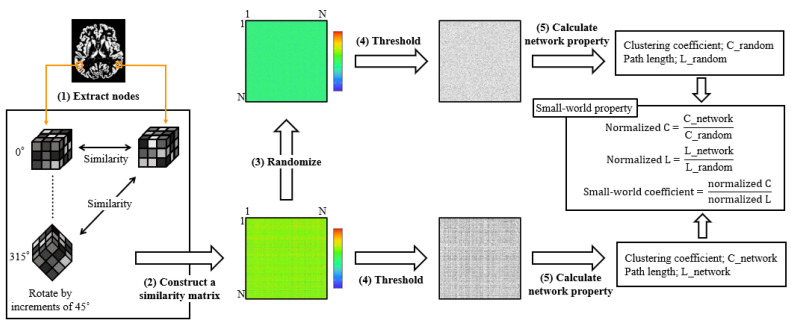
The pipeline of similarity-based gray matter network analysis. After preprocessing, the extracted gray matter image was resliced into small brain regions of 3 × 3 × 3 voxel cubes (1). After the identification of maximum similarity value by rotating each cube with an angle *θ*, the similarity between all the nodes was calculated with the correlation coefficient, and a similarity matrix was constructed (2). Five random matrices were also constructed for each similarity matrix (3). Thereafter, the similarity matrix was binarized with a threshold that ensured a 5% chance of spurious connections for all the participants (4). Finally, we calculated the network properties and small-world properties (5).

**Figure 3 jpm-13-00419-f003:**
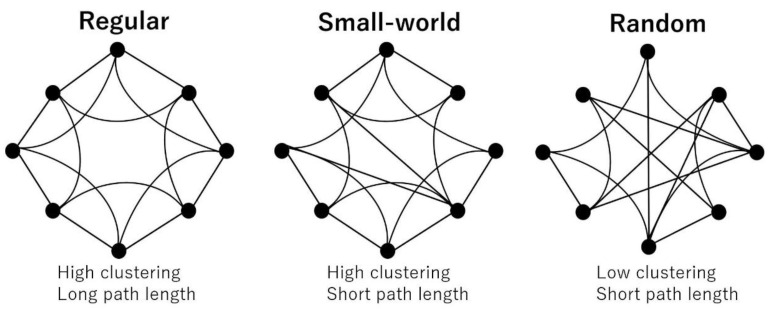
Small-world network model. A regular network has high clustering and long path length, while a random network has low clustering and short path length. A small-world network lies between regular and random, showing high clustering and short path length (many short-range connections coexist with a few long-range connections).

**Figure 4 jpm-13-00419-f004:**
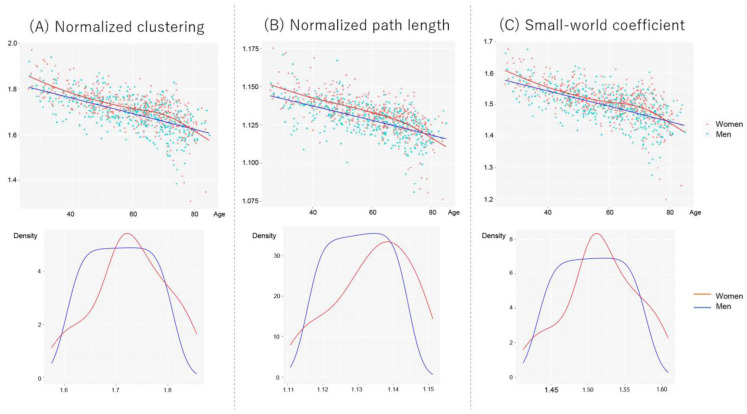
Scatter plots and fit curves of the global network properties for women and men. The fit curves were constructed based on the generalized additive model (GAM). (**A**) Normalized clustering, (**B**) normalized path length, and (**C**) small-world coefficient. Each data distribution for women and men was shown using a kernel approximation.

**Table 1 jpm-13-00419-t001:** Demographic characteristics of the participants.

Characteristic	Women	Men	*p*-Value
Participants, N	407 (50.1%)	405 (49.9%)	
Age, y			
Chronological age	59.272 ± 13.990	59.245 ± 13.970	0.964 ^a^
Predicted brain age	59.019 ± 15.234	58.490 ± 16.186	0.923 ^a^
Brain PAD score	−0.254 ± 5.413	−0.755 ± 5.156	0.139 ^a^
Network measures			
Normalized clustering	1.729 ± 0.086	1.711 ± 0.078	<0.0001 ^b^
Normalized path length	1.112 ± 0.016	1.106 ± 0.015	<0.0001 ^b^
Small-world coefficient	1.554 ± 0.059	1.546 ± 0.056	0.012 ^b^

Data are presented as N (%) and mean ± standard deviation. Brain-PAD score: predicted age-chronological age. ^a^ Mann–Whitney U test. ^b^ Mann–Whitney U test with Bonferroni correction.

**Table 2 jpm-13-00419-t002:** Age- and sex-related network alterations in participants aged < 70 and ≥70 years.

Variables	Unstandardized β	Standard Error	Standardized β	*p*-Value
Chronological age < 70 y				
Normalized clustering				
Age × female (sex) interaction	<−0.001	0.051	−0.045	0.214
Age effect	−0.003	0.037	−0.580	<0.001
Female (sex) effect	0.051	0.065	0.269	0.012
Normalized path length				
Age × female (sex) interaction	<−0.001	0.041	−0.112	0.040
Age effect	<−0.001	0.030	−0.423	<0.001
Female (sex) effect	0.012	0.052	0.262	<0.001
Small-world coefficient				
Age × female (sex) interaction	<−0.001	0.054	−0.014	0.416
Age effect	−0.002	0.039	−0.589	<0.001
Female (sex) effect	0.028	0.069	0.243	0.05
Chronological age ≥ 70 y				
Normalized clustering				
Age × female (sex) interaction	−0.008	0.517	−1.363	0.009
Age effect	0.609	0.603	1.638	0.412
Female (sex) effect	−0.002	0.363	−0.298	0.008
Normalized path length				
Age × female (sex) interaction	−0.001	0.391	−1.223	0.002
Age effect	<−0.001	0.274	−0.162	0.555
Female (sex) effect	0.103	0.456	1.425	0.002
Small-world coefficient				
Age × female (sex) interaction	−0.006	0.555	−1.347	0.016
Age effect	−0.001	0.389	−0.320	0.412
Female (sex) effect	0.421	0.647	1.622	0.015

Data are estimated by a generalized linear model controlling for number of nodes and scanner types.

**Table 3 jpm-13-00419-t003:** Differences in the network properties between women and men in participants with a chronological age < 70 and ≥70 years.

Variables	Women	Men	*p*-Value
Chronological age < 70 yrs			
Normalized clustering	1.756 ± 0.003	1.730 ± 0.003	<0.001
Normalized path length	1.115 ± 0.001	1.110 ± 0.001	<0.001
Small-world coefficient	1.575 ± 0.002	1.558 ± 0.002	<0.001
Chronological age ≥ 70 yrs			
Normalized clustering	1.661 ± 0.009	1.655 ± 0.009	0.679
Normalized path length	1.100 ± 0.001	1.100 ± 0.001	0.873
Small-world coefficient	1.508 ± 0.007	1.504 ± 0.007	0.692

Data are presented as the mean ± standard deviation. Differences between women and men in network properties were analyzed using analysis of covariance, controlling for chronological age, number of nodes, and scanner types.

**Table 4 jpm-13-00419-t004:** Partial correlations of network properties with chronological/predicted brain age in participants < 70 and ≥ 70 years.

	Chronological Age		Predicted Brain Age		Comparisons of Correlations *
	*r*	*p*-Value	*r*	*p*-Value	*p*-Value
<70 yrs					
Normalized clustering	−0.551	<0.0001	−0.590	<0.0001	0.16
Normalized path length	−0.524	<0.0001	−0.573	<0.0001	0.114
Small-world coefficient	−0.526	<0.0001	−0.556	<0.0001	0.233
≥70 yrs					
Normalized clustering	−0.282	<0.0001	−0.628	<0.0001	<0.0001
Normalized path length	−0.269	<0.0001	−0.604	<0.0001	<0.0001
Small-world coefficient	−0.275	<0.0001	−0.612	<0.0001	<0.0001

Correlations between network properties and chronological/predicted brain age were analyzed using partial correlation analysis controlling for sex, number of nodes, and scanner types. * Comparisons of correlations were analyzed using Psychometrica [18].

## Data Availability

All data are available upon reasonable request from the corresponding author.

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
