# Peer review of "Age and Sex-Related Effects on Single-Subject Gray Matter Networks in Healthy Participants"

_jpm, 2023, doi:10.3390/jpm13030419_

Round 1
Reviewer 1 Report
The manuscript “Age and sex-related effects on single-subject gray matter networks in healthy participants” by Shigemoto and colleagues aims at studying sex differences and the impact of age with respect to gray matter networks, by focusing on normalized clustering coefficients, normalized path length, and small-worldness.
The study is interesting, and the methodological flow of the experiment is mostly adequate, although some passages should be improved. The literature and the significance of the study in the current knowledge, however, are insufficiently described in the current version of the manuscript. Please find detailed points below.
· L48-49. It is good that the authors refer to studies which adress the effects of sex and aging on physical features of the brain. However, in the manuscript these studies are often just cited without explaining their results or discoveries. The authors should be more comprehensive and specific in reporting such studies to give a reliable background of their experiment and to appropriately report the existing literature.
To note, the point raised above is valid thoroughout the manuscript (e.g., L57-62). Thus, the authors should extensively revise the text from the introduction to the discussion to improve the quality of the manuscript.
· L83. What does "mean age were balanced" mean? Maybe the authors mean that the age was distributed evenly? Please be clearer and eventually show a histogram/bar graph.
· L112-118. This paragraph is rather unspecific and makes the study non-reproducible.
Please EXTENSIVELY report details for
- the "brain age prediction model"
- the control over the validation procedure
- the PCA analysis
- the method used to predict chronological age.
Please also note that setting the number of PCA to 200 is arbitrary and non-acceptable. Use Monte-Carlo simulation methods like Parallel analysis to estimate the number of PCs to retain.
· With respect to the main results, please report exact p-values. Moreover, report both standardized and unstandardized betas. In general, the tables are confusing with misalignments and missing results (e.g., single effect of age and sex on Normalized path length in Table 2). Please report results according to standards, be clear, and be comprehensive. Also specify what “women / men” means.
· Figure 3. Please also report data distributions (histograms, eventually using a kernel approximation) at least for the Y axis (separate for males and females) in the three plots. This will increase the quantity of information and the graphical quality of the Figure. Moreover, please note that skewed variables may need to be corrected for skewness before statistical tests (e.g., using boxcox methods).
· Please check the reference list for errors. For example, the numbering is wrong (39?)
Reviewer 2 Report
The aim of this study is to highlight age and sex related effects on the grey matter network during healthy normal Aging. The novelty in their methodology comes from utilizing single-subject gray matter brain networks. Below are specific comments on each section.
1) Introduction: The introduction is very well written, a good review of the previous literature and an easy to follow chain of thought leading to the rationale and the proposed experimental plan.
I would recommend adding another paragraph to Elaborate more on the previously identified gray matter Age-related changes and the new insights that a brain network could provide.
2) Material and methods:
In the participant section, the authors reported that the participants have no history of neurological or psychiatric disorders, and no history of 84 use of medication that affects the central nervous system but where there any inclusion exclusion criteria for cardiovascular disease? Masouleh et al 2018 showed evidence for the effect of cardiovascular risk factors on Gray matter structural networks so its worth making sure that any of these isn't a variable that is affecting the results.
3) Results and discussion:
Table 2: In the pValue column, the authors only indicated whether its below or higher than 0.05, an exact pValue would be more sound and consistent with the other data tables.
Other than this the authors interpreted their results clearly, no stretching or over-interpretation was performed. The limitations of the study were highlighted clearly in the discussion section.
Author Response
Reviewer #2
The aim of this study is to highlight age and sex related effects on the grey matter network during healthy normal Aging. The novelty in their methodology comes from utilizing single-subject gray matter brain networks. Below are specific comments on each section.
Reply: Thank you for reviewing our manuscript. We greatly appreciate your helpful comments and suggestions. Please see our point-to-point responses to your comments and a revised manuscript with changes shown in blue.
1) Introduction: The introduction is very well written, a good review of the previous literature and an easy to follow chain of thought leading to the rationale and the proposed experimental plan. I would recommend adding another paragraph to elaborate more on the previously identified gray matter age-related changes and the new insights that a brain network could provide.
Reply: Thank you for your comment. The reviewer has asked to add some sentences concerning about previously identified gray matter age-related changes and the new insights that a brain network could provide. According to the reviewer’s comment, we have added some sentences in the introduction section, as follows. Please note these are shown in red because another reviewer also pointed out.
Page 2 line 52 to line 60 introduction section
“The voxel-based morphometry analysis of 563 healthy subjects (age range: 20-86, female 55%) revealed linear age-related declines of normalized gray matter volume with advancing age [3]. The nonlinear age-related declines of normalized white matter volume (i.e., an inverted U-shape) with slightly increased volume during adulthood which reach a peak in the fourth decade is consistent with the ongoing maturation of the white matter [1]. Regarding the effects of sex, larger normalized gray matter in females compared to males and steeper age-related decline in females compared to males were reported in both younger (142 subjects, age range: 20-34, female 50%) and older groups (135 subjects, age range: 60-86, female 51%) [2].”
Page 2 line 71 to line 76 introduction section
“It has become possible to observe network trajectories related to age based on the individual’s network results, rather than comparing different age groups. Since network properties are linked to cognitive function, this network analysis might provide new information reflecting individual’s brain function. Thus, the careful observation of individual’s network may enable to predict future disease risk”
2) Material and methods:
In the participant section, the authors reported that the participants have no history of neurological or psychiatric disorders, and no history of use of medication that affects the central nervous system but where there any inclusion exclusion criteria for cardiovascular disease? Masouleh et al 2018 showed evidence for the effect of cardiovascular risk factors on gray matter structural networks so its worth making sure that any of these isn't a variable that is affecting the results.
Reply: Thank you for your comment. The reviewer is concerned if there were any inclusion or exclusion criteria for cardiovascular disease. Unfortunately, we didn’t have the information about the cardiovascular risk factors which might affect the structural network results. We greatly thank the valuable comment and would like to make use of our future studies. According to the reviewer’s comment, we have added some sentences in the limitation section, as follows.
Page 10 line 374 to line 379 discussion section
“Fourth, we have no inclusion or exclusion criteria for the risk factors of cardiovascular disease. Previous structural gray matter networks of 616 healthy elderly (age range: 60-80, female 42%) revealed that the cardiovascular risk factors such as smoking, higher blood pressure, higher glucose and higher visceral obesity were negatively associated with structural networks. We have to keep in mind that these risk factors might affect the network declines observed in the elderly in this study.”
Additionally, we added a reference, as follows.
“38. Kharabian Masouleh, S.; Beyer, F.; Lampe, L.; Loeffler, M.; Luck, T.; Riedel-Heller, S.G.; Schroeter, M.L.; Stumvoll, M.; Villringer, A.; Witte, A.V. Gray matter structural networks are associated with cardiovascular risk factors in healthy older adults. J Cereb Blood Flow Metab. 2018, 38, 360-372. DOI: 10.1177/0271678X17729111.”
3) Results and discussion:
Table 2: In the pValue column, the authors only indicated whether its below or higher than 0.05, an exact pValue would be more sound and consistent with the other data tables. Other than this the authors interpreted their results clearly, no stretching or over-interpretation was performed. The limitations of the study were highlighted clearly in the discussion section.
Reply: The reviewer has asked to report exact p-values in Table 2. According to the reviewer’s comment, we have revised Table 2. The reviewer has asked to report exact p-values, both standardized and unstandardized betas, and missing results in Table 2. Because we made the generalized linear model considering the effect of female, therefore, we thought it is better to state age×sex interaction as age×Female interaction and just show age effect and female (sex) effect rather than state the age-related change of women and men. Please see the revised version of table 2 below.
Variables |
Unstandardized β |
Standard error |
Standardized β |
p-value |
Chronological age <70 y |
|
|
|
|
Normalized clustering |
|
|
|
|
Age × Female (Sex) interaction |
<-0.001 |
0.051 |
-0.045 |
0.214 |
Age effect |
-0.003 |
0.037 |
-0.580 |
<0.001 |
Female (Sex) effect |
0.051 |
0.065 |
0.269 |
0.012 |
Normalized path length |
|
|
|
|
Age × Female (Sex) interaction |
<-0.001 |
0.041 |
-0.112 |
0.040 |
Age effect |
<-0.001 |
0.030 |
-0.423 |
<0.001 |
Female (Sex) effect |
0.012 |
0.052 |
0.262 |
<0.001 |
Small-world coefficient |
|
|
|
|
Age × Female (Sex) interaction |
<-0.001 |
0.054 |
-0.014 |
0.416 |
Age effect |
-0.002 |
0.039 |
-0.589 |
<0.001 |
Female (Sex) effect |
0.028 |
0.069 |
0.243 |
0.058 |
Chronological age ≥70 y |
|
|
|
|
Normalized clustering |
|
|
||
Age × Female (Sex) interaction |
-0.008 |
0.517 |
-1.363 |
0.009 |
Age effect |
0.609 |
0.603 |
1.638 |
0.412 |
Female (Sex) effect |
-0.002 |
0.363 |
-0.298 |
0.008 |
Normalized path length |
|
|
|
|
Age × Female (Sex) interaction |
-0.001 |
0.391 |
-1.223 |
0.002 |
Age effect |
<-0.001 |
0.274 |
-0.162 |
0.555 |
Female (Sex) effect |
0.103 |
0.456 |
1.425 |
0.002 |
Small-world coefficient |
|
|
|
|
Age × Female (Sex) interaction |
-0.006 |
0.555 |
-1.347 |
0.016 |
Age effect |
-0.001 |
0.389 |
-0.320 |
0.412 |
Female (Sex) effect |
0.421 |
0.647 |
1.622 |
0.015 |
Round 2
Reviewer 1 Report
The authors did a very good job in reviewing their manuscript.
Thanks and good luck.